# Broadband physical layer cognitive radio with an integrated photonic processor for blind source separation

Weipeng Zhang [1] ✉, Alexander Tait[2], Chaoran Huang[3], Thomas Ferreira de Lima [1,4], Simon Bilodeau[1], Eric C. Blow[1], Aashu Jha[1], Bhavin J. Shastri [5] & Paul Prucnal [1] ✉

The expansion of telecommunications incurs increasingly severe crosstalk and interference, and a physical layer cognitive method, called blind source separation (BSS), can effectively address these issues. BSS requires minimal prior knowledge to recover signals from their mixtures, agnostic to the carrier frequency, signal format, and channel conditions. However, previous electronic implementations did not fulfil this versatility due to the inherently narrow bandwidth of radio-frequency (RF) components, the high energy consumption of digital signal processors (DSP), and their shared weaknesses of low scalability. Here, we report a photonic BSS approach that inherits the advantages of optical devices and fully fulfils its "blindness" aspect. Using a microring weight bank integrated on a photonic chip, we demonstrate energy-efficient, wavelength-division multiplexing (WDM) scalable BSS across 19.2 GHz processing bandwidth. Our system also has a high (9-bit) resolution for signal demixing thanks to a recently developed dithering control method, resulting in higher signal-to-interference ratios (SIR) even for ill-conditioned mixtures.

Many scientific activities, including the earth explorer services[1] (for remote sensing, satellite imagery, radar) and radio astronomy[2], must sensitively detect weak signals at frequencies naturally dictated by physical phenomena (Fig. 1a). Thus, they are vulnerable to radio-frequency (RF) interference from commercial activities whose emission spectrum overlaps with the frequency range of interest. However, this vulnerability to RF interference is increasingly severe because of the expansion of telecommunications. As emerging wireless telecommunication signals[3–5] are squeezed into limited frequency bands, many strategies have been deployed to maximise spectrum utilisation, such as the multi-input and multi-output (MIMO) scheme[6–10], which enhances data-carrying capacity through space-division multiplexing. When many wireless signals transmit simultaneously using frequencies

that are closely spaced, the raised spectral congestion negatively impacts the scientific services through RF interference. This interference issue also poses challenges to the telecom service providers themselves. As the incoming wireless services grant simultaneous access to more users, the tightly packed spatial channels incur severe crosstalk between different users, inevitably degrading the signal quality.

Unlike scientific activities, telecommunication signals are determined a priori. In this case, one way to mitigate spectral congestion is active radio access sharing via cognitive radio[11–13], which dynamically allocates secondary users with access to the unlicensed bands that do not have primary users. This method exploits many spectral gaps between bands licensed by regulatory bodies, which were skipped by

[1]Department of Electrical and Computer Engineering, Princeton University, Princeton 08544 NJ, USA. [2]Department of Electrical and Computer Engineering, Queen's University, Kingston K7L 3N6 ON, Canada. [3]Department of Electronic Engineering, The Chinese University of Hong Kong, Hong Kong, China. [4]NEC Laboratories America, Princeton 08540 NJ, USA. [5]Department of Physics, Engineering Physics & Astronomy, Queen's University, Kingston K7L 3N6 ON, Canada. ✉e-mail: weipengz@princeton.edu; prucnal@princeton.edu

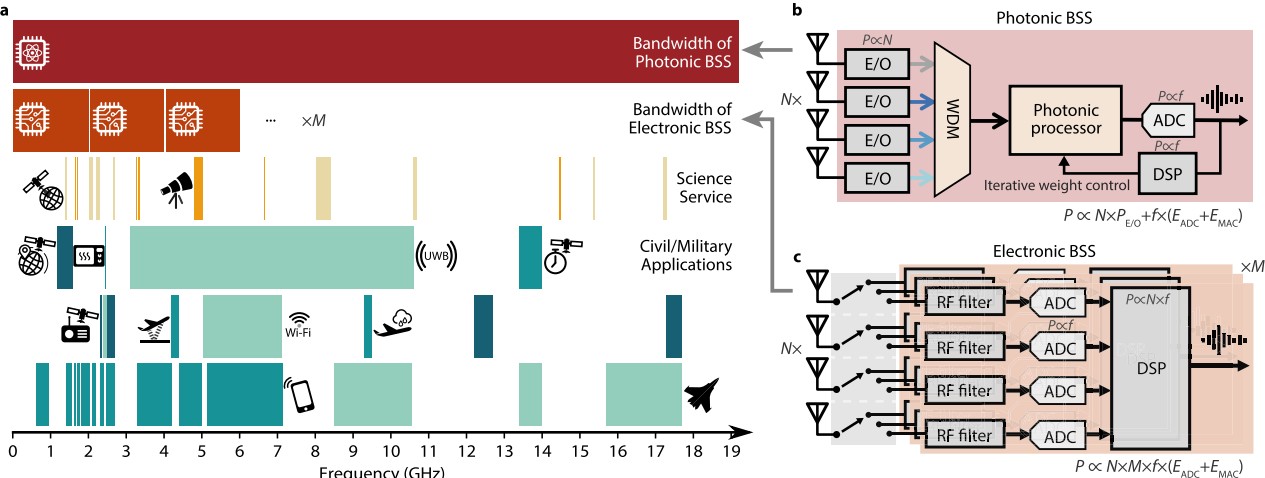

**Fig. 1 | Architecture comparison between electronic and photonic BSS implementation. a** Frequency allocation for common bands (from 0.5 to 19 GHz) and example processing bandwidths for the two BSS implementations (photonics and electronics). Science service bands comprise the earth explorer satellite service (EESS) and radio astronomy service (RAS). Bands for civil and military applications include global positioning system (GPS), microwave oven, ultra-wideband (UWB), and standard frequency and time signal service (for the first row); broadcasting-satellite service, radar altimeter, Wi-Fi, and aircraft weather radar (for the second row); 5 G cellular networks and military radar (for the third row). **b** Photonic BSS system diagram. **c** Electronic BSS system diagram. In electronic BSS, covering the demanded bandwidth requires a complex switching system that consists of multiple electronic BSS setups (quantity of $M$) corresponding to distinct frequency regimes. Each one requires $N$ ADCs ($N$ is the number of receivers) and a dedicated DSP with $N$ inputs to perform the multiply-accumulate (MAC) operation. Since ADC and DSP require power in scale with the signal frequency, the total power consumption of the electronic BSS approaches is proportional to $N \times M \times f \times (E_{ADC} + E_{MAC})$. In contrast, achieving the same coverage requires only one photonic BSS setup consisting of $N$ electrical-to-optical (E/O) converters with only one ADC and a low-end DSP with a single input. This architecture consumes much less power and is proportional to $N \times P_{E/O} + f \times (E_{ADC} + E_{MAC})$.

the existing multiplexing schemes. However, this relies on a complicated software-layer radio-signal identification mechanism susceptible to privacy breaches[14,15]. Substantial signal processing and analysis are required to ascertain whether transmissions are attributable to specific users or others. In situations with many users in wide frequency bands, performing these computations in real time may not be possible. The only feasible way to oversee activity and enforce compliance from a regulatory standpoint may be to record spectral data for offline analysis, which introduces a serious risk to content privacy[16,17]. As soon as information is recorded to disk, its security is compromised. Even if the monitoring operator is deemed benign, it may be unknowingly harbouring malware that can access the content of all the spectrum users.

A physical layer cognitive technique called blind source separation (BSS)[6,18] can extract unknown signals (e.g., a signal of interest and an interferer) from their mixtures with minimal assumptions, as shown in Fig. 1b, c. Operating in the physical layer allows the isolation of unwanted transmissions in the analogue domain, reducing the risk of privacy leakage by eliminating the need to record the transmission content digitally[19]. Such a "blindness" feature is also critically demanded by recovering the scientific signals, for which no prior knowledge can be obtained. Another advantage is the agility in recovering sources with arbitrary characteristics. This means discarding substantial information before digitisation and total obliviousness to the frequency, modulation type, and power ratio. This advantage can only be realised when BSS is performed across a wide frequency range. Conventional BSS implementations by electronics are competent in separating narrow-band and low-frequency signals, such as audio signals[20], but are challenging to achieve a broadband operation due to the limited bandwidth of RF technology. For example, the spectrum of ultra-wideband (UWB) signals[21] covers up to 7.5 GHz, and that of Wi-Fi signals has expanded from 2.4 GHz (802.11) to 6 GHz (802.11ax). Having broadband coverage is challenging with a single RF system, as depicted in Fig. 1. Thus, alternative techniques other than conventional electronic processors are required to effectively process RF signals in next generation wireless systems[22].

By upconverting to frequencies of hundreds of terahertz, photonic signal processors can deal with broadband information[23–25], where GHz signals are regarded as narrowband. As a result, photonic processing can easily satisfy the demanded processing bandwidth requirements of incoming wireless technologies and comes with low energy consumption that does not scale with the signal frequency. A promising on-chip processor is the microring resonator (MRR) weight bank[26,27], a bank of tunable filters implemented with tiny circular optical waveguides, which provides energy-efficient tuning[28] and scalable parallel processing through wavelength-division-multiplexing (WDM). Such an RF frontend powered by a photonic processor (Fig. 1b) can share the workload of signal processing with a digital signal processing (DSP) backend while enhancing overall performance. One key factor determining BSS performance is the resolution of the weights (the tuning accuracy of MRRs), which was reported up to 7 bits on MRRs[29,30]. We recently developed a dithering control method[31] that improves the tuning accuracy beyond 9 bits.

Here, we report a photonic implementation for BSS based on the dithering-controlled MRR weight bank. We also demonstrate a fully packaged photonic processor with a silicon photonic chip integrated with the MRR driver and control electronics on a single printed circuit board (PCB). We prove this setup can recover a weak transmitted signal in the presence of broadband jamming noise and successfully test it in real-time on a wireless transceiver system. In terms of performance, our setup fully realises the "blindness" agility by achieving a processing bandwidth of up to 19.2 GHz and signal-to-interference ratios (SIR) of more than 40 dB in some cases. Besides, the dithering weight control enables a photonic processor with 9-bit accuracy beyond many electronic counterparts, enhancing the BSS with at least one-half reduced residual error than the setup without the dithering control. This work introduces a functional BSS system capable of operating at broad bandwidths. When included in transceiver circuits, it can help cancel interference signals, with potential implications in next-generation wireless cognitive radio to deal with signals across tens of Gigahertz. It also benefits radio astronomy in detecting unknown weak

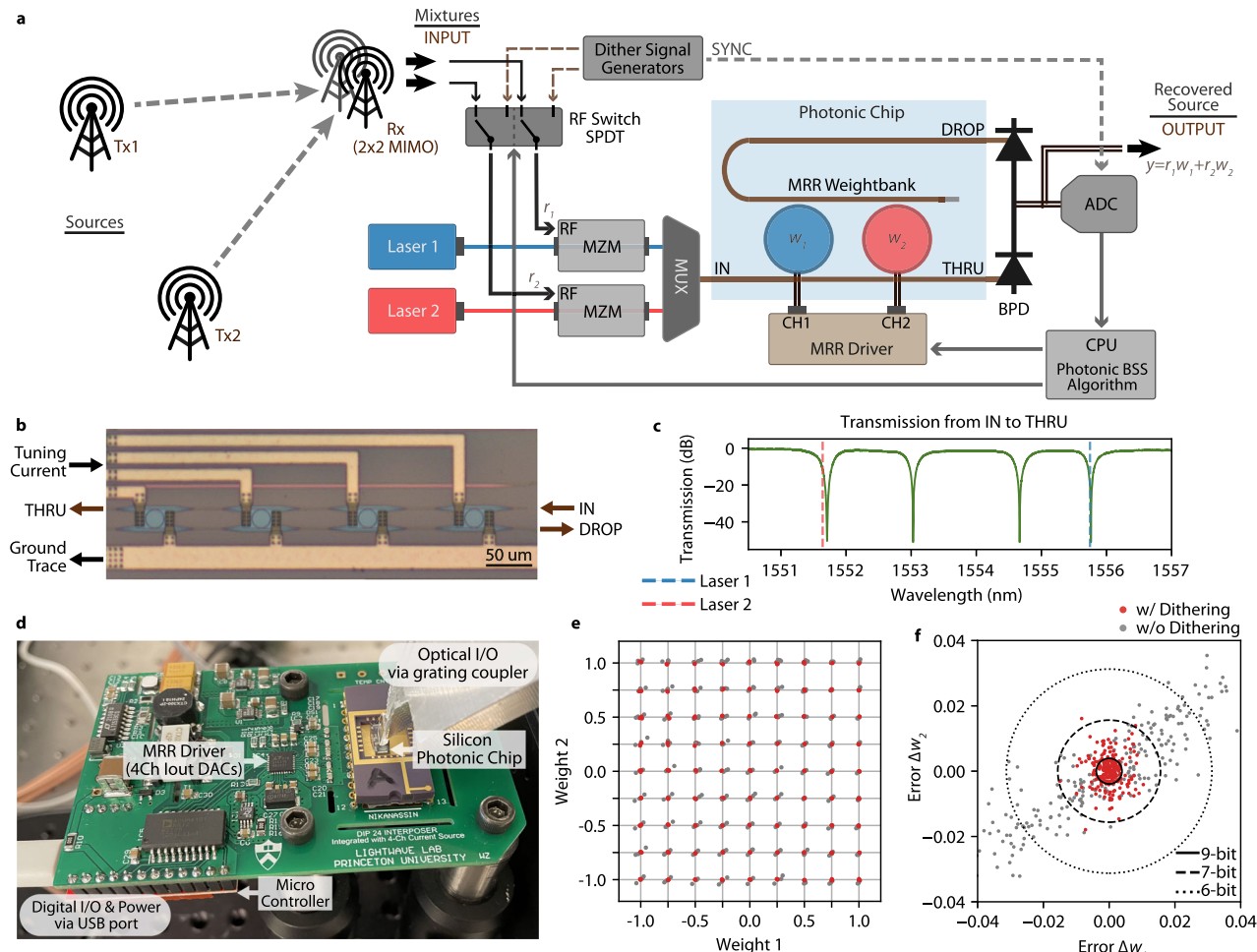

**Fig. 2 | Photonic BSS experimental setup. a** Schematic of the BSS setup. MZM, Mach-Zehnder modulator. MUX, wavelength-dependent multiplexer. BPD, balanced photodetector. ADC, analogue to digital converter. Tx, transmitter. Rx, receiver. **b** Micrograph of the MRR weight bank on the chip. **c** Transmission spectrum of the THRU port at 25 ˚C. **d** Fully packaged photonic processor. The silicon photonic chip, wire-bonded on a DIP 24 chip carrier, is mounted on the right side of this PCB. A 4-channel current output DAC is integrated on the broad and applies currents for tuning the MRRs. Optical input and output (I/O) are through the grating coupler on the top right, and electrical I/O is set up via a USB cable (bottom left), which also delivers the power for the whole PCB. **e** Estimation of weighing accuracy. The 2-MRR weight bank in **a** was tested to tune the weights represented by each grid. The dithering control yielded the red points, and the grey points were obtained without the dithering. **f** Errors of all the tested weights in **d**. A 9.0-bit of precision resulted from the dithering control and 6.7-bit for the control without the dithering. The precision is calculated by the standard deviation of the error. Quantitatively, there are 161 among the total 243 tested points inside the 9-bit bound (the solid circle).

signals that require more than 30 dB signal-to-noise and distortion ratio (SINAD).

## Results

### Photonic solution for BSS problem

The BSS problem can be formulated as separating an unknown mixture of unknown independent signals. The instantaneous model of generic mixing in a transmission channel is $[r_1, r_2] = \mathbf{H}[s_1, s_2]$, where $(s_1, s_2)$ are the source signals, $\mathbf{H}$ is a mixing matrix representing the wireless channel, and $(r_1, r_2)$ are the received signals. In the most general case, the source signals and their mixing matrix are unknown to the receiver. The goal of BSS is to estimate the corresponding demixing matrix, $\mathbf{H}^{-1}$, and apply it to the received signals to arrive at $[s'_1, s'_2] = \mathbf{H}^{-1}[r_1, r_2]$, where $(s'_1, s'_2)$ is the estimated recovery of the source signals. In general, minimal prior assumptions are needed about signal characteristics. For example, $s_1$ and $s_2$ can occupy the same frequency band, meaning an implementation based on filtering would fail regardless of any receiver-side analysis. It is assumed that the two sources are statistically independent and that all mixing happened linearly. These

assumptions are very realistic and are widely used in the radio community[32].

For a given mixing matrix $\mathbf{H}$, separating each signal component requires the mixtures to be weighted and summed with weights represented by each column of the inverse matrix $\mathbf{H}^{-1}$. This operation can be implemented with a set of two MRR weight banks acting as an on-chip signal processor. Retaining the signal of interest and eliminating the other one utilises one of the two columns, which requires only one weight bank. As shown in Fig. 2a, c, the MRR weight bank consists of several microring resonators with slightly different radii; each has a Lorentzian-shaped transmission profile (as shown in Fig. 2c) centred at different wavelengths. Each MRR is equipped with a metal heater to allow thermo-optic displacement of the centre wavelength by varying the current applied[33]. The MRR weight bank independently weights the laser amplitudes of different wavelengths. The sum of all optical power can be obtained by a balanced photodetector (BPD) at the output. Utilising this ability of weighted addition, we develop a photonic BSS algorithm, which follows a pipeline consisting of three steps, including principal component analysis (PCA)[34], whitening, and

independent component analysis (ICA) (See details in ref. [35]). For carrying out these analyses, an iterative algorithm is preferred because of its simplicity in that only one vector needs to be commanded to the MRR weight bank in each step. Essentially, a constrained Nelder-Mead algorithm[36] is carried out that performs an iterative projection-pursuit of the mixtures to search the optimised weighting vectors. The goal is to find the mixture that outputs a weighted addition ($\Sigma w_i s_i, w_i \in [-1,1], i \in [1,2,\cdots N]$, $N$ is the number of the mixtures) with maximal variance (the second-order statistic) for PCA and the maximal non-Gaussianity (the fourth-order statistic or kurtosis) for ICA.

## Hardware implementation

The hardware realisation of this algorithm appears as a control loop (as shown in Fig. 2a). Apart from the photonic chip, also included are a BPD for electrical-to-optical (E/O) conversion (Discovery semiconductor DSC-R405ER), an analogue-to-digital converter (ADC) for signal digitisation (Tektronix DPO73304SX; sampling rate: 625 MS s$^{-1}$–100 GS s$^{-1}$), a computer for statistical analysis and actuating weight commands, and a multi-channel current source for MRR tuning (custom-built as shown in Fig. 2d). The sampling rate of the ADC is set to the minimum of 625 MS s$^{-1}$ for all the tested signals as the BSS method is agnostic to waveform frequencies, and it is switched to the maximum of 100 GS s$^{-1}$ for recording resulted waveforms with the highest definition. The dithering control[31] implemented here allows tuning the MRRs with less complicated drivers instead of the source-measurement unit (SMU)[29]. In this setup, the MRR driver is directly integrated into the PCB interposer, packaged close to the photonics chip with a much-reduced footprint and cable management[31]. The signal path starts from the Mach-Zehnder Modulator (MZM) and ends at the scope. The highest supported RF frequencies are determined at up to 20 GHz by the BPD and the transmission profile of the MRRs, providing coverage for many commonly used RF bands. Detailed discussion on the bandwidth of the MRR filtering function can be found in Supplementary Fig. 2. It is also worth noting that most of the signal path is in the optical domain, bringing about broadband and flat response and very low latency, which is estimated to be 15 ns by dividing the total waveguide length (3 m) by the speed of light in the waveguide ($c_0/n \approx 2 \times 10^8$ m s$^{-1}$).

The photonic chip in this setup has a 4-channel MRR weight bank with resonance frequencies roughly spaced by 200 GHz. The spectra of the four MRRs (at 25 degrees) are shown in Fig. 2c, with the resonance peaks located at 1551.7, 1553.0, 1554.6, and 1555.7 nm. The waveguide of these MRRs is N-doped which can be efficiently thermal-tuned from on to off positions with a power consumption of 10 mW[28]. Since this work recovers source signals from two mixtures, we use two MRRs (the leftmost and rightmost ones). The corresponding lasers (Pure Photonics PPCL500) are tuned to be 1551.5 and 1556 nm, then amplified (Pritel FA-23) and combined into a shared waveguide by a WDM multiplexer (Santec MDM-15-8) before coupling into the MRR weight bank through grating couplers.

The implemented dithering control method overcomes the low accuracy incurred by the high sensitivity of the weight bank. As shown in Fig. 2a, the lasers are modulated with either the received mixtures or pre-defined dithering signals. Each time a set of commanded weights are applied, the RF switch (Mini-Circuits RC-2SPDT-A26; DC − 26.5 GHz) passes the dithering signals into the photonic path, which helps adjust the driving currents of the MRRs until the output weights reach the demanded values. Then, the actual mixtures are switched into the weight bank and processed. The weight accuracy is reflected in the error between the target and actual weights. Usually, we quantify the accuracy in bits, which is calculated as $\log_2(2/(w_{\text{actual}} - w_{\text{target}}))$. Figure 2e, f illustrates the weighting accuracy, showing the resulting weights (red dots) of the two MRRs being examined at the tested values represented by each grid point. The grey dots correspond to the same targeted weight without dithering control. An improvement of

over 2 bits (from 6.7 to 9 bits) is observed, enabling the MRR weight bank to have competitive performance with its electronic counterparts.

While our setup included the demonstration of the wireless transceiver system, we also had a versatile control setup that provided flexibility and accuracy in controlling the carrier frequencies and the mixing matrix. In the control setup, the signal mixtures were generated by a high-speed multi-channel arbitrary waveform generator (Keysight N8196A; 92 GS s$^{-1}$) and sent to each MZM directly. The generation of the two baseband signals, the up-conversion, and the signal mixing, are all performed via software tools (Python). The mixed signals are then transmitted to the photonic system.

## Theoretical impact of weighting accuracy

BSS must be able to deal with mixtures that are difficult to separate. Denoting the $j$th signal component in the $i$th mixture as $y_{ij}(t)$, the signal-to-interference ratio (SIR)[18] is defined as

$$\text{SIR}_i(\text{dB}) = 10\log_{10}\frac{||y_{ij'}(t)^2||}{\Sigma_{j \neq j'}||y_{ij}(t)^2||} \quad (1)$$

which is the ratio of the signal power ($||y_{ij'}(t)^2||$) to the rest interference power ($\Sigma_{j \neq j'}||y_{ij}(t)^2||$) of the $i$th mixture. Given a problem with $N$ mixtures containing $N$ sources to be separated, an often-used merit is the overall SIR, which is the average of the SIR of every mixture ($1/N \times \Sigma_i \text{SIR}_i$, $i = 1, 2, \cdots N$). The SIR captures the accuracy of the system, and it can also be stated as a function of carrier frequency, baseband bandwidth, or other metrics. A higher SIR means better suppression of the interference signals. This metric of SIR can be extended to any mixing $\mathbf{H}$ through the ill-condition number, $\kappa(\mathbf{H})$[37], defined as Eq. (2).

$$\kappa(\mathbf{H}) = ||\mathbf{H}|| \cdot ||\mathbf{H}^{-1}|| \quad (2)$$

This describes the demixing difficulty calculated by the mixing matrix $\mathbf{H}$. Mixtures with a small ill-condition number are easier to solve. Conversely, problems with a sizeable ill-condition number are challenging and prone to smaller SIR. An ill-conditioned BSS problem typically requires the weighting to represent the inverse matrix accurately.

To prove this, consider a simple case where the mixing matrix is symmetric, such that $\mathbf{H} = [[1, 1 - a], [1 - a, a]]$, $a \in [0.5, 1]$, which is often the case when two receiver antennas and the two transmitter sources are symmetrically positioned and have identical power. The inverse of matrix $\mathbf{H}$ is given in Eq. (3).

$$\mathbf{H}^{-1} = \frac{a}{2a-1}\begin{bmatrix} 1 & (a-1)/a \\ (a-1)/a & 1 \end{bmatrix} \quad (3)$$

And this can be further normalised to the form of Eq. (4), multiplying $(2a-1)/a$.

$$\mathbf{H}'^{-1} = \begin{bmatrix} 1 & (a-1)/a \\ (a-1)/a & 1 \end{bmatrix} \quad (4)$$

To introduce the weighting error, Eq. (5) describes the actual matrix of mixtures that is input to the photonic BSS, where $d$ denotes the weight error caused by the inaccurate MRR control.

$$\mathbf{H}''^{-1} = \begin{bmatrix} 1+\delta & (a-1)/a+\delta \\ (a-1)/a+\delta & 1+\delta \end{bmatrix} \quad (5)$$

$$[s_1', s_2'] = \mathbf{H}''^{-1}[r_1, r_2]^T = \mathbf{H}''^{-1}\mathbf{H}[s_1, s_2]^T = \begin{bmatrix} 1+\delta & (a-1)/a+\delta \\ (a-1)/a+\delta & 1+\delta \end{bmatrix}\begin{bmatrix} s_1 \\ s_2 \end{bmatrix} \quad (6)$$

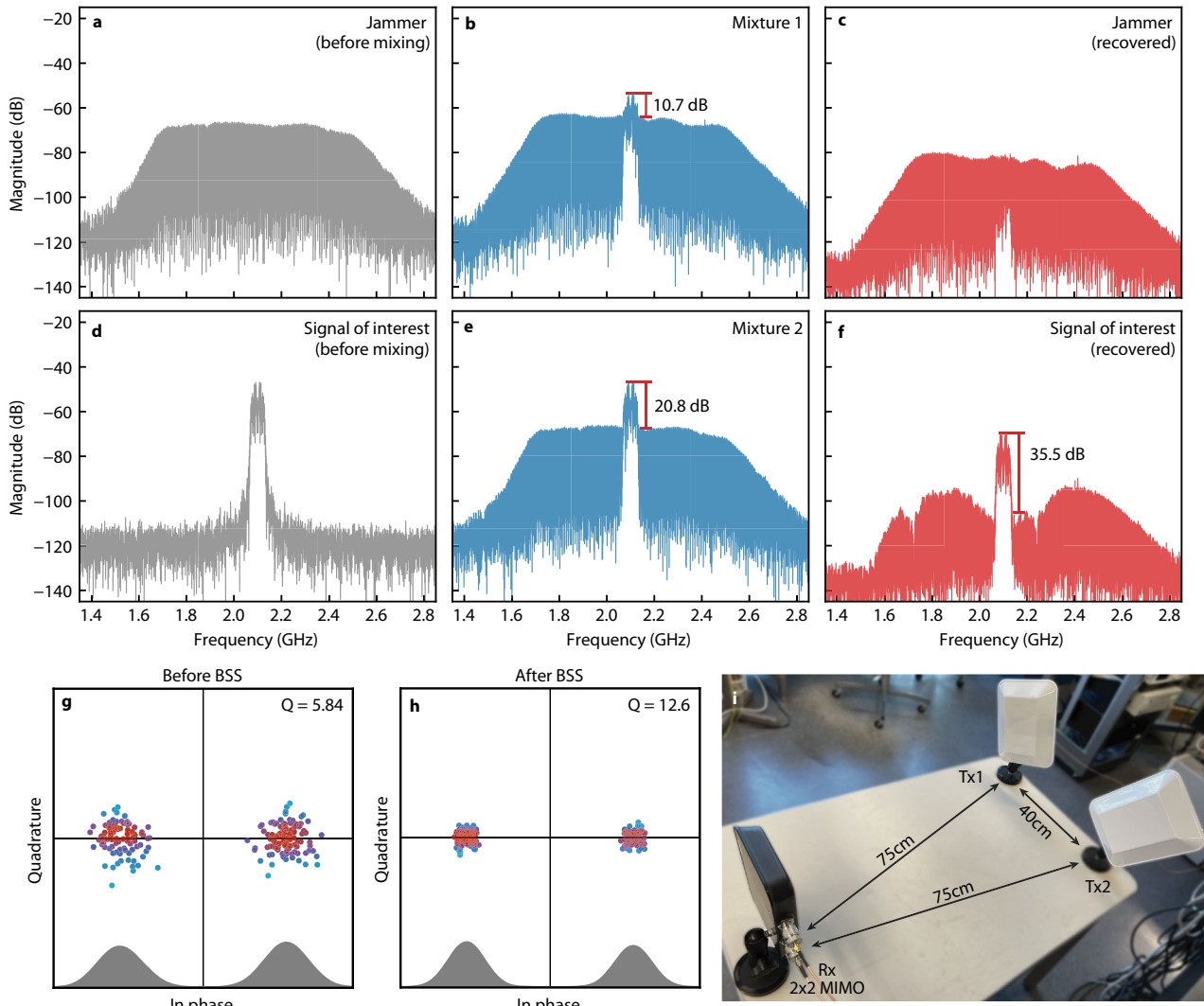

**Fig. 3 | BSS demonstration on wireless transceiver system. a, d** Spectrum of the source (Tx1) and the jammer (Tx2). These were measured at the receiver when one of the two transmitters (Tx1 or Tx2) was tuned off. **b, e** Spectrum of received mixtures. **c** Spectrum of the separated jamming noise. **f** Spectrum of the recovered source. **g, h** Constellation diagram of the received mixture (**g**, also corresponding to **e**) and recovered source (**h**). The colour represents the density. The $Q$ values are 1.90 and 5.84 for the mixtures corresponding to **b** and **e**, respectively. **i** Antenna setup.

The coefficients (proportional to the amplitudes of each source signal) of the separated results can then be expressed by Eq. (6); the percentage of error (ignoring the noise) that remains in the recovered results can be represented as $a\delta/(2a+2a\delta-1)$. Thus, if $a=0.9$ as given in this problem, the control accuracy improvement from 6.7 bits ($\delta=2/2^{6.7}\approx0.019$) to 9 bits ($\delta\approx0.004$) could, in theory, lower the error by a factor of 4.6 (from 2.0% to 0.44%). With these equations, we can also derive that if 2% error (SIR≈33) is demanded, this 2.3-bit improvement in control accuracy will increase the solvable ill-condition number from 2.05 ($a=0.9$) to 10 ($a=0.55$).

**Demonstration on wireless transceivers**
Based on the setup described above, we firstly demonstrated our proposed BSS photonic processor in a wireless transceiver system, which emulates the case where a communication link is deteriorated by nearby RF interference. As shown in Fig. 3i, two antennas (Southwest Antennas 1009-002; 1.7-2.5 GHz) transmit the signal of interest and a broadband jammer mixed over the air with a transmission distance of 0.75 m. Then, the mixtures are received by a 2 × 2 MIMO antenna (Southwest Antennas 1055-368; 1.7–2.5 GHz), with two outputs corresponding to the polarisation of 45-degree

slant left and 45-degree slant right. The signals emitted from the transceivers have peak-to-peak voltages of 1 V and received signals are 150 and 140 mV peak-to-peak. The transmitted signal carries a repeating sequence of 200 random bits at a baud rate of 50 MHz with a binary phase-shift keying (BPSK) modulation format and a carrier frequency of 2.1 GHz. The interference is an instantaneous broadband jamming signal generated by adding 10,000 single tones with random phases and has a spectrum from 1.7 to 2.5 GHz, covering the entire bandwidth of the antennas. Thus, extracting the signal of interest is largely ineffective via spectral filtering even if the carrier frequency is known. If doable, the signal-to-noise ratio remained the same with the received mixtures at best. In contrast, photonic BSS can recover the communication link without prior assumptions and suppress the interference noise. Figure 3a–h illustrates the spectrum and the constellation diagram before and after the BSS process. The signal-to-noise ratio has almost a 15 dB improvement (20.8–35.5 dB), accompanied by a twofold increase of the Q value (5.84–12.6) from the constellation. This result demonstrates the effective suppression of nearby interference with high power and broadband coverage, maintaining the transmission quality of the wireless communication link.

## Broadband capability

Next, we examined the proposed photonic BSS system under different scenarios, with different signal mixtures enabled by programming the arbitrary waveform generator (AWG). The original two signals are repeating patterns of 16 bits, and are in format of binary phase-shift keying (BPSK; bit pattern = [0,1,0,1,0,1,0,1,0,1,0,1,0,1,0,1]) and on-off keying (OOK; bit pattern = [0,0,1,0,0,0,1,0,0,1,0,0,0,1,0,0]), respectively. This configuration provided two short data periods that can be easily illustrated and also contributed all the potential bits combinations in their mixtures for a complete examination. Note that our BSS algorithm is based on the kurtosis analysis, independent of the modulation format (see Methods and Supplementary Fig. 1 for details). For consistency, all the experiments shown in the main content were performed on signals in the same digital modulation formats (BPSK or OOK). For reference, we performed an additional experiment showing similar BSS performance on analogue modulated signals (pulsed ultra-wideband signal) in Supplementary Fig. 3. Based on this setup, we firstly tested the broadband capability by performing BSS on mixtures of signals from 20 MHz to 19.2 GHz by varying the carrier frequencies, as shown in Fig. 4. The baseband frequencies were also adjusted according to the carrier frequencies, which were 4 MHz for <80 MHz, 16 MHz for 80–480 MHz, 160 MHz for 0.48–1 GHz, 400 MHz for 1–3 GHz, 800 MHz for 3–6 GHz, and 1600 MHz for $f_{carrier} \geq 6.4$ GHz. In the spectrum, the transmitted signals are centred close to the carrier frequency with a width (instantaneous bandwidth) of approximately double the baseband. Annotating the two mixtures with $M_1$ and $M_2$ and the two sources with $S_1$ and $S_2$, the mixing can be expressed as $M_1 = 0.8 \times S_1 + 0.2 \times S_2$ and $M_2 = 0.2 \times S_1 + 0.8 \times S_2$, denoting an ill-condition number of 2.26 (according to Eq. (2)). The two mixtures are identical in power, with both the peak-to-peak voltages set to 1 V.

As shown in Fig. 4d, the 27 tested carrier frequencies from 20 MHz to 19.2 GHz show SIR >30 dB. This result proves that our demonstrated photonic BSS system can process RF waveform with a carrier frequency lower than 19.2 GHz and a baseband of up to 1.6 GHz (an instantaneous band of 3.2 GHz). Compared with previous photonic demonstration[35] that dealt with problems of a similar ill-condition number, we obtain almost 55 times broader processing bandwidth (19.2 GHz versus previously 350 MHz centred at 900 MHz) while maintaining a clean signal separation across the entire band (SIR > 30 vs previously SIR≈14). This improvement in error suppression confirms the benefit of the improved dithering control method. Also, based on the Federal Communications Commission (FCC) frequency allocation chart (partly shown in Fig. 1a), this broadband coverage by a single silicon photonic chip translates into the agility of processing multiple commonly used bands. Examples of included bands are cellular (620 MHz–6.425 GHz), Wi-Fi (2.4 GHz, 5–7.125 GHz), military radar (extensive spectral usage above 8.5 GHz), and those for earth explorer satellite and radio astronomy (sparsely spread from 1.4–17.3 GHz). This wide processing bandwidth can also provide full coverage to some challenging bands, such as the ultra-wideband (UWB) services. Since signals remain narrowband at higher frequencies for photonic devices and the state-of-the-art BPD can be 100 GHz or more[38], this system can easily expand the coverage to other important spectrums like millimetre-wave with just higher speed photodetectors. Due to the shape of the spectral profile (as shown in Fig. 2c), MRR weight banks apply uneven filtering on signals with large instantaneous bands, which could degrade the BSS performance to some extent. Additional discussion and experiments are included in Supplementary Information, where we show that this BSS system maintains decent performance (SIR ≥ 33 dB) on two ultra-wideband (UWB) signals with a 7.5 GHz wide instantaneous spectral coverage (3.1–10.6 GHz). The practical device footprint on-chip is 0.13 mm × 0.42 mm, including four MRRs and the waveguide routing, which slowly scales up linearly with the number of sources.

## Test on ill-conditioned mixing

Last, we investigated the performance of the proposed photonic BSS system in solving problems with different ill-condition numbers, which is to justify the significance of the improved weighting accuracy. Here, we fixed the carrier frequency at 1 GHz. The mixing matrix is of a symmetric form $\mathbf{H} = [[a, 1 - a], [1 - a, a]]$, where $a$ is varied from 0.1 to 0.45, resulting in ill-condition numbers ranging from 2.05 to 10.1. Figure 5 shows the SIR of the mixtures obtained from the same setup but with the dithering control (compared to the previous control method without the dithering). Even in the presence of similar experimental noise levels, lower SIR is always obtained when not using dithering control. Conversely, the dithering controlled setup maintained constantly high SIR such that the influence of ill-conditioning was less distinguishable. The average SIR is above 35 dB for all tested cases, which generally shows around 20 dB improvement compared to the previous control method (the orange curve in Fig. 5), as expected in the analysis above. This improvement confirms the significance of accurate weight control for MRR-based applications, such as the BSS in this paper.

## Discussion

In summary, we explored a physical layer cognitive radio solution based on BSS performed on a dithering-controlled silicon MRR weight bank. This solution is a complete RF frontend with a photonic signal processor that can do intelligent learning through the fully programmable and integrated electronic-photonic system. This setup has a large processing bandwidth of up to 19.2 GHz that can fully demonstrate the capability of the BSS technique. In addition, the high SIR observed for all the frequencies and ill-conditioned problems, together with an example of a wireless transceiver system, confirms the benefits of real-world applications brought by the improved MRR control method. Given the increasing carrier frequencies in the next generation of wireless communications technologies, the superior performance of this photonic approach illustrates the readiness to replace pure-electrical RF implementations, effectively addressing the incoming challenges, including bandwidth limitations, energy efficiency, and latency.

With the availability of higher speed modulators[39] and photodetectors[38] on the silicon platform, as well as the maturity of packaging, including photonic wire bonding[40], laser integration[41], and monolithic cointegration of complementary metal–oxide–semiconductor (CMOS) and silicon photonics[42], this proposed photonic BSS can have future implementations with higher integration and broader bandwidth. With these, we envision a standalone BSS device of a small form factor that is field-deployable for various applications, including interference cancellation in autonomous vehicles and aviation.

## Methods

### Photonic BSS algorithm

The photonic BSS algorithm follows a pipeline of two steps, including principal component analysis (PCA) and independent component analysis (ICA). Firstly, PCA is to find the principal components (PC) from the received mixtures and construct a whitening matrix. In PCA, we use the MRR weight bank to perform a weighted addition of the received mixtures. Regarding all the applied weights as a vector, the PCA is to find the vector that the corresponding output weighted addition has the maximal possible variance. Finding the PC vectors is through the constrained Nelder-Mead (NM) method[43]. Starting from a random vector, the NM method iteratively tries new vectors and updates the old one if the new vector results in a larger variance. The new vectors are generated by manoeuvring the old ones in several different ways, including reflection, expansion, shrinking, etc. Usually, the search converges within 20 steps and takes no more than 5 minutes. Denoting this founded PC as the first PC, the vector

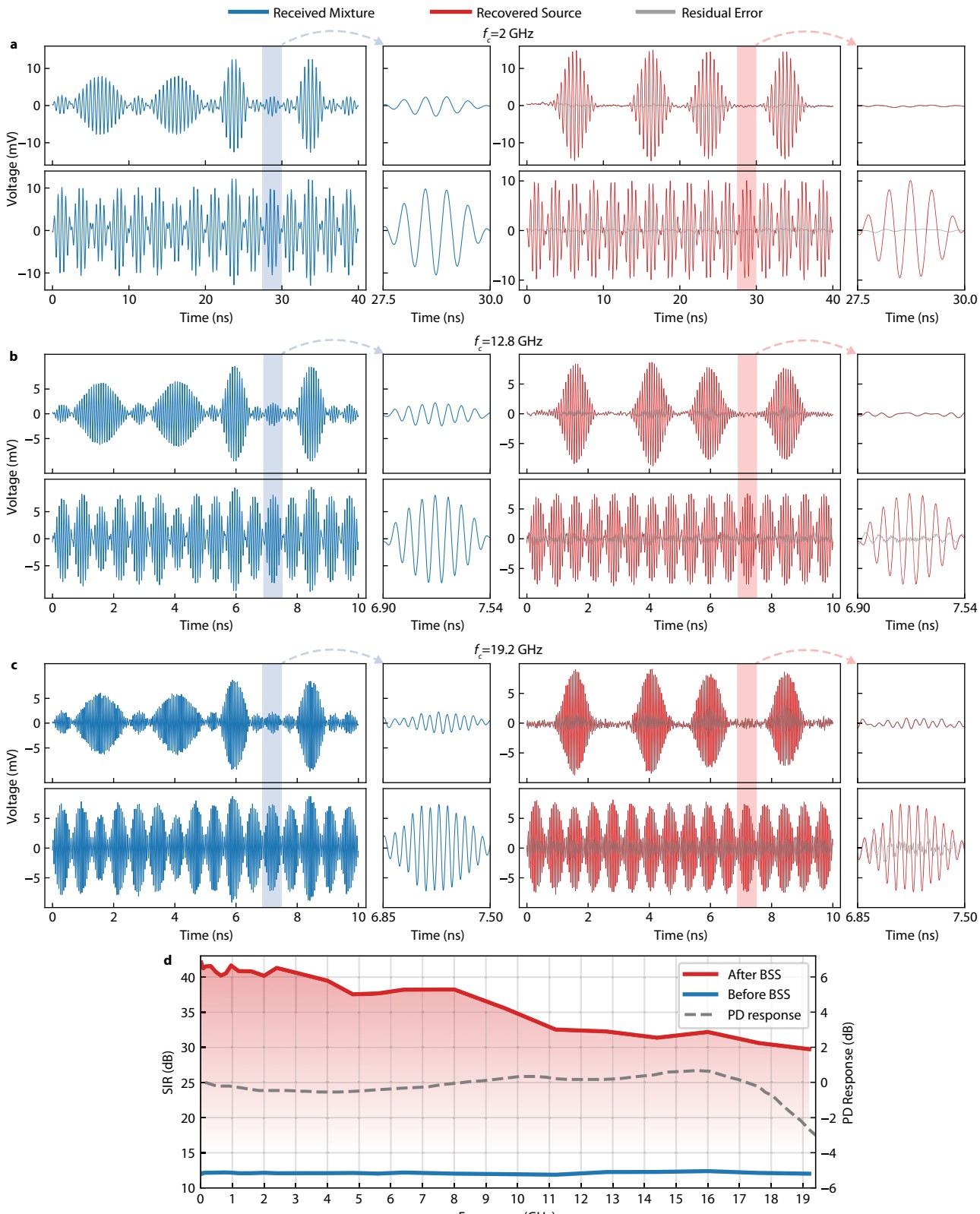

**Fig. 4 | Broadband BSS with a single photonic chip. a−c** BSS results, mixtures (blue) and estimated sources (red) with carrier frequencies of 2, 12.8, and 19.2 GHz, respectively. The residual errors are in grey. **d** SIR of the received mixtures (blue curve) and the recovered sources (red curve) versus the carrier frequency. The dashed grey curve is the relative power response of the photodetector.

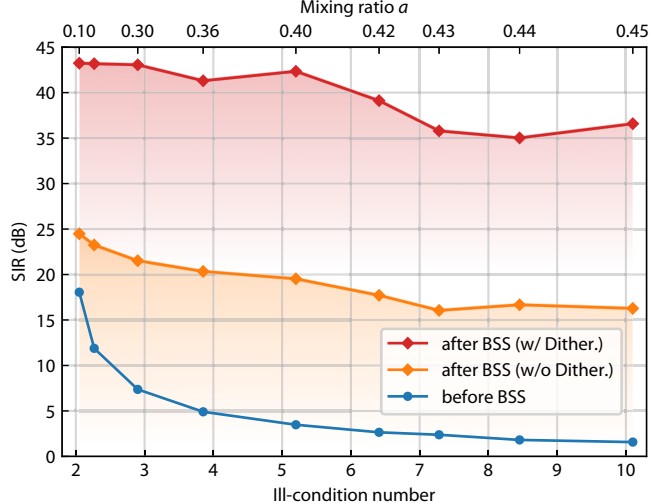

**Fig. 5 | BSS performance on ill-conditioned mixtures.** Blue, red, and orange curves are the SIR before the BSS, after BSS with dithering control, and after BSS without dithering control, respectively. The mixing matrix **H** is defined by the mixing ratio $a$ by $\mathbf{H} = [[a, 1-a], [1-a, a]]$. The corresponding ill-condition number is calculated by Eq. (2).

perpendicular to the first PC should associate with the minimum variance and is called the second PC. So, finding the first one can easily derive the other by a $\pi/2$ shift. With this, a whitening matrix can be constructed using the two PCs and their associated variances. Secondly, the ICA step follows a similar search procedure and finds the vector (denoted as the first independent component (IC)) corresponding to the maximal kurtosis. Based on the central limit theory, the mixture of two uncorrelated signals is more Gaussian-like than any original signals. The kurtosis is a metric describing the non-Gaussianity. Thus, the vector obtained by ICA is exactly one of the correct demixing weights that can recover the one original signal from the mixtures. Finally, using the first IC and the whitening matrix obtained in the PCA step, the second IC can be directly calculated and used to recover the second original signal. This BSS pipeline does not require explicit waveform digitisation but only the second-order and fourth-order statistics (variance and kurtosis) of weighted addition output, unlike conventional BSS solutions such as FastICA. This feature permits a low-cost ADC and DSP working at the sub-Nyquist sampling regime, generally more applicable to broadband BSS. Detailed pseudo codes regarding the algorithm mentioned above can be found in our previous works[34,44].

### Fully packaged photonic processor
We fully packaged the silicon photonic chip with its driver and control circuitry in the same interposer PCB in our experimental setup, as shown in Fig. 2d. This integration benefits a simplified lab setup, lower power consumption (<100 mW), and neat connectivity that a single USB cable handles both the digital interface and the power delivery. In terms of power management, the 5 V input (from the USB cable) generates isolated and low-noise power rails by several dedicated power management chips (Analog Devices LT1533CS and LT3042). These rails power up the precision current sources (Analog Devices LTC2662-16; 16-bit), which drive the MRRs on the photonic chip. The digital interface between the current source and the controller (Sparkfun Arduino Pro Micro) is built on the serial peripheral interface (SPI) protocol and isolated by a dedicated chip (Analog device ADUM4151). This fully isolated MRR driving circuity helps with noise suppression. The host computer commands the weights through serial communication via the USB cable. The onboard controller phrases the

received command and talks to the current sources to adjust the current of each microring.

### Device fabrication
The silicon photonic chip was fabricated on a silicon-on-insulator wafer with a silicon thickness of 220 nm and a buried oxide thickness of 2 μm. The waveguide is 500 nm wide. The weight bank consists of four MRRs (radius around $r = 22$ μm) coupled with two bus waveguides in an add/drop configuration, and two among the four were used in the experiments. A slight difference ($\Delta r = 0.32$ μm) was introduced in the ring radii to avoid resonance collision. The gap between the ring and bus waveguide is 200 nm, yielding a Q factor of about 6000. Circular metal heaters were built on top of each MRR for thermally weight tuning. Metal vias and traces were deposited to connect the heater contacts of the MRR weight bank to electrical metal pads.

### Data availability
The source data generated in this study have been deposited in the Figshare database under accession code https://doi.org/10.6084/m9.figshare.21670880.

### Code availability
All codes used in this study are available from the corresponding authors upon request.

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

## Acknowledgements

This research is supported by the National Science Foundation (NSF) (ECCS-2128616 to P.P. and ECCS-1642962 to P.P.), the Office of Naval Research (ONR) (N00014-18-1-2297 to P.P. and N00014-20-1-2664 to P.P.), and the Defense Advanced Research Projects Agency (HR00111990049 to P.P.). The devices were fabricated at the Advanced Micro Foundry (AMF) in Singapore through the support of CMC Microsystems. B. J. Shastri acknowledges support from the Natural Sciences and Engineering Research Council of Canada (NSERC). S. Bilodeau acknowledges funding from the Fonds de recherche du Québec - Nature et technologies.

## Author contributions

W.Z., T.F.L. and A.T. conceived the ideas and implemented the experimental setup, designed the experiment, conducted the experimental measurements, and analysed the results. T.F.L., E.C.B., S.B. and A.J. designed the silicon photonic chip. T.F.L., C.H., A.T. and B.J.S. provided the theoretical support. E.C.B. and S.B. performed the chip packaging. W.Z., A.T., C.H., S.B. and B.J.S. wrote the manuscript. P.P. supervised the research and contributed to the general concept and interpretation of the results. All the authors discussed the data and contributed to the manuscript.

## Competing interests

The authors declare no competing interests.
