## [Peer Review File · Nature Communications]

Broadband physical layer cognitive radio with an integrated photonic processor for blind source separationREVIEWER COMMENTS

Reviewer #1 (Remarks to the Author):

In this manuscript, the authors proposed a BSS method that was performed on a dithering controlled silicon MRR weight bank. As the authors demonstrated, the proposed system is a complete RF frontend with a photonic signal processor that can do intelligent learning through the fully programmable and integrated electronic-photonic system. The proposed system and BSS algorithm covered a large operation frequency range of 19.2 GHz, about 55 times of the traditional electric processing bandwidth. I think this work is innovative and the results are encouraging. It is nice work that fully reflects the advantage promote practical applications of photonic signal processing techniques. This manuscript has very clear expression about the principle, results and discussion. I think it can be considered for publication. There are two suggestions for the authors.

1. It is not rigorous to use “bandwidth” when describing the “carrier frequency range”. The “carrier frequency” and “signal bandwidth” should be distinguished.

2. The comparison of SIR improvement by using the proposed photonic and traditional electrical methods is not sufficient. Typically, the performance of an electric system would be deteriorated obviously as the carrier frequency becomes larger. While, the photonic system usually have nearly the same performance over a very large operation frequency range. Considering the power loss and nonlinearity due to the E/O and O/E conversions, it is expected that, the traditional electrical BSS method could achieve better performance over the photonic, when the carrier frequency is low. The authors only emphasized the SIR improvement in high frequency range and large bandwidth. The conditions for low carrier frequency should also be considered.

Reviewer #2 (Remarks to the Author):

The submitted paper presents a Blind Source Separation system that exploits the wide bandwidth of photonics to allow the frequency- and bandwidth-agnostic operation. The work is very interesting: it illustrates a significant device that could be further developed into a product allowing improving performance of RF transmission systems. Moreover, the work is strongly multi-disciplinary, since it deals with integrated photonics and signal processing, and it has also requested photonic packaging as well as electronics capabilities.

Nevertheless, several points could be better described to simplify the read and strengthen the paper:

- The motivations for the approach focus on the application to scientific activities such as Earth observation and radio astronomy. Is there any reason why these applications are considered as the target ones, and not the other RF communications systems?

Moreover, radars and radio astronomy typically use analog signals, while the experimental test on the presented device has considered digital modulations. In case an analog signal would be used in the system, would it be there any difference in the device performance?

By the way, the second paragraph of the paper (from line 56 on) discusses about cognitive radio for the management of resources. This sounds a bit useless when considering radars or radio astronomy signals, for which it is not possible to manage resources: the used frequencies are determined a-priori.

Radars require very large SINAD (signal to noise and distortion ratio), in the order of several tens of dB. The performance reported in the paper reveals an SNR of 35dB. It could be useful to state which kind of application would benefit of such results. For example, radars for Earth observation should actually require about 30dB...

- The photonic approach is said to allow wideband operation, and this has been demonstrated in the chapter “Broadband capability” in the sense that the central frequency of the signal of interest has been changed across an ultra-wide frequency range from 1 to 19.2GHz.

But the analysis has been run with signals with 50MHz instantaneous bandwidth.

What would happen to signals with much broader bandwidth (as UWB services)? Would it there any effect of noise build-up?

Similarly, which is the bandwidth of the MRR filtering function?

- Also, the method is said to be power efficient, but no details on the power consumption are reported. At least, the consumption of the MRR controls should be given.

Similarly, the latency is not discussed.

- Personally, I found hard to follow the paragraph "Photonic BSS algorithm" in the Methods. Although this is not my area of expertise, I believe that few more explanations could benefit the non-expert reader. Moreover, I think it could be useful to know the sampling rate at the oscilloscope: this would allow understanding the convenience of the waveform-agnostic approach.

- The power of the signals is not defined.

- In Fig.2f, not all the red points show >9 bits. How have you calculated the precision? Is that an average of the collected points?

- In the "Results" paragraph, the reported model of generic mixing considers two signals. In the case study, the target is to isolate one signal from an interferer. Is it correct to consider the interferer as a signal, knowing that it could be the random summation of several transmission channels with different frequencies, bandwidths, statistics?

- SIR at line 111 is not defined (the definition of the acronym comes later in the paper). PC and IC at lines 350 and 356 are not defined neither.

- In "Photonic hardware implementation", the external MZMs are not described.

- The position along the text of Eq. (3) to (5) is strange. It would be better having the equations in the middle of the phases that cite them. Moreover, I found difficult following the calculations (I admit I am not an expert of these technical details).

- At line 283, the mixing of the two signals is given, I believe one of the two should be M2...

Point-by-point reply to reviewers' comments:

Reviewer 1

In this manuscript, the authors proposed a BSS method that was performed on a dithering controlled silicon MRR weight bank. As the authors demonstrated, the proposed system is a complete RF frontend with a photonic signal processor that can do intelligent learning through the fully programmable and integrated electronic-photonic system. The proposed system and BSS algorithm covered a large operation frequency range of 19.2 GHz, about 55 times of the traditional electric processing bandwidth. I think this work is innovative and the results are encouraging. It is nice work that fully reflects the advantage promote practical applications of photonic signal processing techniques. This manuscript has very clear expression about the principle, results, and discussion. I think it can be considered for publication. There are two suggestions for the authors.

- 1) *It is not rigorous to use "bandwidth" when describing the "carrier frequency range". The "carrier frequency" and "signal bandwidth" should be distinguished.*

Response: Thank you for this suggestion. We acknowledge the ambiguity between "signal bandwidth" and "carrier frequency." To clarify, the signal bandwidth is often related to the frequency coverage of the baseband signal, while the carrier frequency is the frequency used for upconverting the baseband signal. Thus, in the spectrum, the transmitted signals are centered close to the carrier frequency with a frequency width of approximately double the baseband. In this work, our proposed photonic BSS approach is successfully demonstrated on signals with carrier frequencies of up to 19.2 GHz and an instantaneous band of 3.2 GHz (the baseband is 1.6 GHz).

In our setup, the carrier frequency's limiting factor is the photodetector's frequency response, which can be extended in the future with improved components (as discussed in the conclusion section of the manuscript). Besides, the transmission profile of the MRR also raises limitations in processing signals with a broad instantaneous band, that the slope of the profile can introduce uneven spectral filtering. In the added Supplementary Information, we theoretically conclude that this uneven filtering insignificantly degrades the suppression of unwanted signals but can distort the signal of interest. For most applications where signals are of a narrow instantaneous band (<1 GHz), the distortions caused by MRRs are moderate. Besides, the distortions are of good linearity and can be compensated with a simple equalizer when processing broadband signals. We provide additional experimental proof by demonstrating this BSS system on mixtures of signals (ultra-wideband) with an instantaneous bandwidth of 7.5 GHz. The signal-to-interference (SIR) result is as good as 33 dB, which is the same performance as the narrowband signals shown in the main article.

We changed the terms used in the revised manuscript to remove any ambiguity. We adopted "processing bandwidth" to refer to the spectral coverage of the signals that can be processed. We also added descriptions to clarify how the range of carrier frequency of the tested signals can relate to the "processing bandwidth" of our proposed photonic BSS system. The revisions include the following:

[Original] "...The SIR captures the accuracy of the system, and it can also be stated as a function of frequency, bandwidth, or other metrics..." (Line 217)

[Revised] "...The SIR captures the accuracy of the system, and it can also be stated as a function of carrier frequency, baseband bandwidth, or other metrics..." (Line 220)

[Original] "...and 1600 MHz for $f_{carrier} \geq 6.4$ GHz..." (Line 281)

[Revised] "...and 1600 MHz for $f_{carrier} \geq 6.4$ GHz. In the spectrum, the transmitted signals are centered close to the carrier frequency with a width (instantaneous bandwidth) of approximately double the baseband..." (Line 292)

[Original] "...As shown in Fig. 4d, the 22 tested frequencies from 1 GHz to 19.2 GHz show SIR of no less than 30 dB." (Line 285)

[Revised] "...As shown in Fig. 4d, the 27 tested carrier frequencies from 20 MHz to 19.2 GHz show SIR greater than 30 dB. This result proves our demonstrated photonic BSS system can process RF waveform with a carrier frequency of lower than 19.2 GHz and a baseband of up to 1.6GHz (an instantaneous band of 3.2GHz) ..." (Line 300)

[Original] "...this system can easily expand the coverage to other important spectrums like millimeter-wave just by using higher speed photodetectors..." (Line 300)

[Revised] "...this system can easily expand the coverage to other important spectrums like millimeter-wave with just higher speed photodetectors. Due to the shape of the spectral profile (as shown in Fig. 2c), MRR weight banks apply uneven filtering on signals with large instantaneous bands, which could degrade the BSS performance to some extent. Additional discussion and experiments are included in the Supplementary Information, where we show that this BSS system maintains decent performance ($SIR \geq 33$ dB) on two ultra-wideband (UWB) signals with a 7.5 GHz wide instantaneous spectral coverage (3.1 - 10.6 GHz)..." (Line 316)

2) *The comparison of SIR improvement by using the proposed photonic and traditional electrical methods is not sufficient. Typically, the performance of an electric system would be deteriorated obviously as the carrier frequency becomes larger. While, the photonic system usually have nearly the same performance over a very large operation frequency range. Considering the power loss and nonlinearity due to the E/O and O/E conversions, it is expected that, the traditional electrical BSS method could achieve better performance over the photonic, when the carrier frequency is low. The authors only emphasized the SIR improvement in high frequency range and large bandwidth. The conditions for low carrier frequency should also be considered.*

Response: Thank you for bringing this to our attention. We agree with the reviewer about the performance comparison between photonic and electrical methods when dealing with signals of low frequencies. There, electrical methods can perform better thanks to lower noise, higher sensitivity, lower energy consumption,

etc. However, due to the narrow processing bandwidth of electrical systems, achieving the same spectral coverage as our proposed photonic system would require using multiple electrical BSS systems, each targeting a specific frequency range, and a complex switching system to integrate across these bands. Because of this enormous hardware budget, we think a direct performance comparison between electrical and photonic methods using metrics such as the SIR would not be accurate.

However, to quantify the performance of our photonic method at lower frequencies, we have performed additional experiments and added significant information to the revised manuscript. For frequencies down to 20MHz (shown in Fig. 4), we confirm that our system achieves high performance regarding SIR, which is more than 40 dB at the lower frequencies. We also mention in the introduction that electrical systems are potentially better options for processing low-frequency signals, such as audio signals. We have added additional references to support this point. We have revised the conclusion with discussions on the advantages of a photonic approach, over purely electrical, for the next generation of wireless communications technologies with increasing carrier frequencies and signal bandwidth. The revisions include the following:

[Original]

(Fig. 4d)

[Revised]

(Fig. 4d)

[Original] "...Based on this setup, we firstly test the broadband capability by performing BSS on mixtures of signals from 1 GHz to 19.2 GHz by varying the carrier frequencies, as shown in Fig. 4. The baseband frequencies were also adjusted according to the carrier frequencies, which were 160 MHz for less than 1 GHz, 400 MHz for 1 - 3 GHz, 800 MHz for 3 - 6 GHz, and 1600 MHz for $f_{carrier} \geq 6.4$ GHz..." (Line 279)

[Revised] "...Based on this setup, we firstly test the broadband capability by performing BSS on mixtures of signals from **20 MHz** to 19.2 GHz by varying the carrier frequencies, as shown in Fig. 4. The baseband frequencies were also adjusted according to the carrier frequencies, which were **4 MHz for less than 80 MHz, 16 MHz for 80 - 480 MHz, 160 MHz for 0.48 - 1 GHz**, 400 MHz for 1 - 3 GHz, 800 MHz for 3 - 6 GHz, and 1600 MHz for $f_{carrier} \geq 6.4$ GHz..." (Line 287)

[Original] "...As shown in Fig. 4d, the 22 tested frequencies from 1 GHz to 19.2 GHz show SIR of no less than 30 dB." (Line 285)

[Revised] "...As shown in Fig. 4d, the **27** tested **carrier** frequencies from **20 MHz** to 19.2 GHz show SIR greater than 30 dB. ..." (Line 300)

[Original] "...However, a broadband operation can be challenging for the electronic implementations of BSS due to the limited bandwidth of conventional RF technology..." (Line 82)

[Revised] "...**Conventional BSS implementations by electronics are competent in separating narrowband and low-frequency signals, such as audio signals²⁰**, but are challenging to achieve a broadband operation due to the limited bandwidth of RF technology..." (Line 81)

[Original] "...The superior performance of this photonic approach illustrates the readiness to replace conventional RF implementations, effectively addressing the incoming challenges in wireless communications, including bandwidth limitations, energy efficiency, and latency." (Line 334)

[Revised] "...**Given the increasing carrier frequencies in the next generation of wireless communications technologies**, the superior performance of this photonic approach illustrates the readiness to replace **pure-electrical** RF implementations, effectively addressing the incoming challenges, including bandwidth limitations, energy efficiency, and latency." (Line 351)

Reviewer 2

The submitted paper presents a Blind Source Separation system that exploits the wide bandwidth of photonics to allow the frequency- and bandwidth-agnostic operation. The work is very interesting: it illustrates a significant device that could be further developed into a product allowing improving performance of RF transmission systems. Moreover, the work is strongly multi-disciplinary, since it deals with integrated photonics and signal processing, and it has also requested photonic packaging as well as electronics capabilities. Nevertheless, several points could be better described to simplify the read and strengthen the paper:

- 1) *The motivations for the approach focus on the application to scientific activities such as Earth observation and radio astronomy. Is there any reason why these applications are considered as the target ones, and not the other RF communications systems?*

Response: We thank the reviewer for raising this question. The motivation of our work is the combination of BSS—which is oblivious to signal format and data rate—and the photonic approach, which enables broad bandwidth operation. This may solve RF interference issues in many scenarios. Scientific activities, including earth exploration, radio astrometry, and so on, are an especially well-suited application since they involve detecting unknown signals at sparse target frequencies spread across tens of gigahertz within their spectrum. Conventional electrical solutions cannot offer this coverage within a single device or at a manageable size, weight, and power (SWaP). This is not to say that our photonic approach is limited to applications like recovering scientific signals since it is also beneficial in places where broad bandwidth is critical and conventional electrical solutions are lacking. This is not uncommon in newly developed wireless technologies, such as ultra-wideband used for accurate positioning, as well as Wi-Fi 6 and 5G/6G networks that boost the communication speed manyfold. All these are achieved by extending the spectrum usage. Besides the higher data rates, another feature of next-generation wireless technology is higher capacity granting access to more users simultaneously. As a result, the same spectral band can be squeezed with signals from different service providers (ISPs) in various formats. This introduces the challenge for base stations or edge users of RF interferences between unknown signals spread widely within the spectrum. Hence, our proposed photonic BSS can efficiently address this issue with superior SWaP.

We have revised the introduction, highlighting the advantages of implementing BSS on silicon photonic chips and elevating its significance to applications other than scientific activities. We hope this can give a better insight to potential readers on the advantages of the photonic approach and provide a complete picture of what photonic BSS is capable of.

[Original] "...The inevitable downside incurred is degraded signal quality due to severe crosstalk among tightly packed spatial channels. Also, when many wireless signals transmit simultaneously using frequencies that are close together, the raised spectral congestion negatively impacts the scientific services through RF interference..." (Line 51)

[Revised] "...When many wireless signals transmit simultaneously using frequencies that are close together, the raised spectral congestion negatively impacts the scientific services through RF interference. This interference issue also poses challenges to the telecom service providers themselves. As the incoming wireless services grant simultaneous access to more users, the tightly packed spatial channels incur severe crosstalk between different users, inevitably degrading the signal quality..." (Line 51)

[Original] "...As a result, photonic processing has low energy consumption that does not scale with the signal frequency." (Line 92)

[Revised] "...As a result, photonic processing can easily satisfy the demanded processing bandwidth for next-generation wireless technologies and comes with low energy consumption that does not scale with the signal frequency...." (Line 91)

2) *Moreover, radars and radio astronomy typically use analog signals, while the experimental test on the presented device has considered digital modulations. In case an analog signal would be used in the system, would it be there any difference in the device performance?*

Response: This is an interesting insight. Indeed, there are many important applications utilizing analog signals. For example, radar systems commonly use short pulses or frequency-swept signals, which are similar but differ from the digital modulations widely used in communications.

BSS is known to be agnostic to the signal format because its algorithm does not rely on explicit and prior knowledge about the waveform. As described in the Methods in the manuscript, we utilized independent component analysis (ICA), which follows the central limit theorem and can find the demixing weights that output a signal with the optimized kurtosis. The central limit theory purports that the mixtures of two independent signals should be more Gaussian in amplitude distribution than any original signals before mixing. The kurtosis is a metric associated with the signal distribution; it is calculated by the second and fourth statistic moment of signals using Equation 1 below, where \bar{s} is the mean of the signal.

$$\text{Kurtosis}(s(t)) = \frac{E[(s(t)-\bar{s})^4]}{(E[(s(t)-\bar{s})^2])^2} \quad (1)$$

A perfect gaussian signal has a kurtosis of 3. Non-gaussian distributed signals can have a kurtosis either larger than or smaller than 3. Thus, an often-used metric to describe the non-gaussianity is the absolute relative kurtosis (ARK), which is calculated by $\text{ARK} = \text{abs}(\text{kurtosis}(s(t)) - 3)$ for a given signal, $s(t)$. The search for optimal demixing weights is done through a constrained Nelder-Mead algorithm. This is a non-gradient-based optimization algorithm that approaches the maximal ARK (maximal non-gaussianity) by iteratively updating a searching simplex, or in our case, the weights applied onto the on-chip microring resonators.

Hence, our approach does not distinguish signals from their modulation format, either digital or analog. There is no direct mathematical evidence on which modulation format has a larger ARK. Fig. R1 below displays some standard signals of both modulation types, suggesting that signals with digital modulation are usually associated with smaller ARKs. For an iterative search procedure like the Nelder-Mead algorithm, a target associated with a smaller output of the objective function (that is, the ARK in the BSS problem) is more difficult to find. Such scenarios often require more iterative steps to converge, or yield less accurate weights, leading to signals that are not optimally separated. Therefore, separating digitally modulated signals, as is shown in the manuscript, is a more challenging task to examine the performance of this photonic blind source separation. In addition, we demonstrate our BSS system for separating analog signals in the Supplementary Information, where two pulse-shaped ultra-wideband (UWB) signals are separated from their mixtures with the same performance as the digital ones, as shown in Fig. 4 in the manuscript.

Fig. R1. Kurtosis of standard analog and digital modulated signals. These signals share the same carrier frequency of 1 GHz. **a-c** analog signals that are colored blue color. **d,e** digital modulated ones colored in orange. **a1-e1** the histogram corresponding to signals shown in **a-e**, respectively.

The significant revisions include the following:

[Original] "...This configuration provided two short data periods that can be easily illustrated and also contributed all the potential bits combinations in their mixtures for a complete examination..." (Line 274)

[Revised] "... This configuration provided two short data periods that can be easily illustrated and also contributed all the potential bits combinations in their mixtures for a complete examination. **Note that our BSS algorithm is based on the kurtosis analysis, independent of the modulation format (see Methods and Supplementary Information for details). For consistency, all the experiments shown in the main content were performed on signals in the same digital modulation formats (BPSK or OOK). For reference, we performed an additional experiment showing similar BSS performance on analog modulated signals (pulsed ultra-wideband signal) in the Supplementary Information...**" (Line 280)

Additional revisions are included in sections 1 and 3 of the **Supplementary Information**.

3) *By the way, the second paragraph of the paper (from line 56 on) discusses about cognitive radio for the management of resources. This sounds a bit useless when considering radars or radio astronomy signals, for which it is not possible to manage resources: the used frequencies are determined a-priori. Radars require very large SINAD (signal-to-noise and distortion ratio) in the order of several tens of dB. The performance reported in the paper reveals an SNR of 35dB. It could be useful to state which kind of application would benefit of such results. For example, radars for Earth observation should actually require about 30dB...*

Response: Thank you for these suggestions. We agree that cognitive radio is less advantageous in applications where the signal of interest is unknown, such as radars and radio astronomy. Our photonic method is experimentally tested to bring about more than 30 dB SNR, which, as pointed out, is demanded by radars where large SINADs are critical. Besides these applications, the broadband capability of this photonic approach makes it also suitable for wireless communication applications like cellular and Wi-Fi communications. These signals can be known before being transmitted, and cognitive radio technology is an effective approach in this scenario. As discussed in the manuscript, cognitive radio suffers from complicated mechanisms and privacy threats, which the photonic system does not. To avoid confusion and make our point clearer to readers, we made several changes, such as adding the SINAD requirement by astronomy radars and stating that our photonic approach can be used and benefit such applications. Besides, we clarify the application scope of where the cognitive radio method can be utilized. We believe these clarifications better highlight the significance of our photonic approach.

[Original] "...One way to mitigate spectral congestion is active radio access sharing via cognitive radio..." (Line 56)

[Revised] "...**Unlike scientific activities, telecommunication signals are determined a-priori. In this case,** one way to mitigate spectral congestion is active radio access sharing via cognitive radio..." (Line 55)

[Original] "...Another advantage of "blindness" is the agility in recovering sources with arbitrary characteristics..." (Line 77)

[Revised] "... Such a "blindness" feature is also critically demanded by recovering the scientific signals, for which no prior knowledge can be obtained. Another advantage is the agility in recovering sources with arbitrary characteristics..." (Line 75)

[Original] "...When included in transceiver circuits, it can help cancel interference signals, with potential implications in cognitive radio and radio astronomy." (Line 116)

[Revised] "...When included in transceiver circuits, it can help cancel interference signals, with potential implications in next-generation wireless cognitive radio to deal with signals across tens of Gigahertz. It also benefits radio astronomy in detecting unknown weak signals that require more than 30 dB signal-to-noise and distortion ratio (SINAD)." (Line 114)

4) *The photonic approach is said to allow wideband operation, and this has been demonstrated in the chapter "Broadband capability" in the sense that the central frequency of the signal of interest has been changed across an ultra-wide frequency range from 1 to 19.2GHz. But the analysis has been run with signals with 50MHz instantaneous bandwidth. What would happen to signals with much broader bandwidth (as UWB services)? Would it there any effect of noise build-up? Similarly, which is the bandwidth of the MRR filtering function?*

Response: We welcome this constructive comment. Let us first discuss the bandwidth of the MRR filtering function. The resonator-based optical filters, such as the MRRs, achieve weighting of signals by tuning their transmission spectrum (as shown in Fig. 2c and Fig. R2 below). Each MRR in the weight bank filters the light from the input port into the thru and drop port. The transmission of the optical power is shown in Fig. R2a. Then, a balanced photodetector (BPD) converts the optical power to electrical signals in proportion to the power difference between the drop and thru ports, as shown in Fig. R2c. In our experiment, we used the two MRRs whose resonance frequencies are centered around 192700 GHz and 193200 GHz, corresponding to the leftmost and the rightmost peaks, respectively. We used two lasers operating in frequencies within the slopes (the orange and the green regions) and modulated the RF signals onto the lasers. Then, these two MRRs filtered their corresponding laser light to the drop and thru ports, and the resulting electrical output from BPD is the weighted summation of the two input signals. As shown in Fig. R2b and d, both positive and negative weights can be applied to carry out the BSS algorithm and suppress the unwanted signal.

Fig. R2. MRR Spectra. **a** the transmission of the MRR weight bank. The blue and red curves are for the drop and thru ports, respectively. The four transmission peaks and dips correspond to four independent MRRs inside the weight bank, as shown in Fig. 2b. Noting that the y-axis here is in linear scale as opposed to the log scale used by Fig. 2c. **b-d** the weight as a function of the laser frequency. **b** and **d** are the zoomed-in views of the orange and the green regions in **c**, respectively. The curve in **b** is centered at 192666 GHz, and that in **d** is at 193229 GHz.

We acknowledge that the linewidth of the laser is widened when modulating RF signals onto the lasers. Neglecting the linewidth of the unmodulated laser, the widened linewidth is double the maximal signal frequency, which is approximately the carrier frequency plus half the signal bandwidth. Zooming in on the weighting curves (shown in Fig. R2b and c), we can notice that the slope of this profile can theoretically introduce non-constant filtering onto the signal if either the carrier frequency or the instantaneous bandwidth is excessively large. Potentially causing distortions for processed signals, which partially explains why the SIR performance is decreased as the carrier frequency increases, as depicted in Fig. 4. However, the incurred distortions are mitigated by themselves. The transmission profiles of these two MRRs are of very similar Lorentzian shapes because of their almost identical ring radii. As shown in Fig. R2 b-d, we used different sides of the transmission profiles so that the filtering function from one MRR is positively sloped (Fig. R2b) and the other is negatively sloped (Fig. R2d). Thus, the uneven frequency responses derived from the slopes are canceled to some extent when the weighted signals are summed at the BPD, which maintains a good suppression of signals with broad bandwidth or large carrier frequencies.

Besides, the slope is associated with the transmission profile width, and intentional narrow widths were designed to avoid spectrum overlaps and the potential crosstalk between adjacent MRRs. Suppose fewer MRRs are needed, like only two of the four MRRs were required for the demonstrated BSS experiment, the distortion can be further mitigated by optimizing the profile. The transmission profile can be controlled by the device geometry, including the radius of the MRRs and the distance between the main and the ring waveguide.

In the manuscript, we prove this with experimental results showing more than 30 dB SIR for signals with carrier frequencies of up to 19.2 GHz and bandwidth of 3.2 GHz. In the Supplementary Information of the revised manuscript, we added additional experimental results using this photonic BSS system separating two UWB signals (3.1-10.6 GHz) to test our claims further. The results prove similar performance, i.e., SIRs of more than 33 dB, to those shown in Fig. 4. This demonstrated performance is sufficient for most wireless applications that utilize small portions of the spectrum guided by FCC allocation.

In terms of noise, there lacks a particular relation between the noise and the instantaneous bandwidth of the signals. The photodetector has a frequency-dependent response, as represented by the dashed curve in Fig. 4d, which is decently flat within a broad frequency range. The dominant noise in our system should be the shot noise associated with the photodetector, which is related to the optical power and has no direct formula with the signal frequency. Thus, we think our proposed system should process the instantaneous broadband signals with the same performance in terms of noise. The BSS demonstration on UWB signals supports this claim, details of which are included in the Supplementary Information.

We hope these discussions will help provide a clearer picture to the readers. We made several adjustments to the revised manuscripts: We have highlighted the instantaneous bandwidth of the signals used in the results of Fig. 4. We added the above discussion in the Supplementary Information. Based on the same setup of Fig. 4, we performed another experiment using this photonic BSS system separating two UWB signals. All the new information in the Supplementary Information is cross-referenced in the manuscript for ease of reference.

[Original] "...this system can easily expand the coverage to other important spectrums like millimeter-wave just by using higher speed photodetectors..." (Line 300)

[Revised] "...this system can easily expand the coverage to other important spectrums like millimeter-wave just by using higher speed photodetectors. **Due to the shape of the spectral profile (as shown in Fig. 2c), MRR weight banks apply uneven filtering on signals with large instantaneous bands, which could degrade the BSS performance to some extent. Additional discussion and experiments are included in the Supplementary Information, where we show that this BSS system maintains decent performance (SIR \geq 33 dB) on two ultra-wideband (UWB) signals with a 7.5 GHz wide instantaneous spectral coverage (3.1 - 10.6 GHz)...**" (Line 316)

[Original] "...The signal path starts from the MZM and ends at the scope, and the highest supported RF frequencies are determined by the BPD of up to 20 GHz, providing coverage for many commonly used RF bands." (Line 171)

[Revised] "...The signal path starts from the MZM and ends at the scope, and the highest supported RF frequencies are determined at up to 20 GHz by the BPD and **the transmission profile of the MRRs,**

providing coverage for many commonly used RF bands. **Detailed discussion on the MRR filtering function can be found in the Supplementary Information.**" (Line 170)

Additional revisions are included in sections 2 and 3 of the **Supplementary Information**.

5) *Also, the method is said to be power efficient, but no details on the power consumption are reported. At least, the consumption of the MRR controls should be given. Similarly, the latency is not discussed.*

Response: We agree that power consumption and latency are important performance parameters that should be highlighted in the manuscript. The MRR we used is thermally tuned. When applying current to the ring's N-doped waveguide through metal traces and vias built on the chip, the waveguide heats, which changes the silicon's index of refraction, shifting the ring's transmission profiles in the optical spectrum. Ring resonator actuation is very sensitive and energy efficient because a tiny change in the index of refraction is sufficient to achieve the required spectral shift, leading to a calculated tuning efficiency of 28 mW for tuning the MRR transmission across a free-spectral range (FSR). In practice, 10 mW power is sufficient for tuning the MRR from "on-state" to "off-state." We omit the overall power consumption since it is not a straightforward measurement and is less meaningful for readers given the variety of setup implementations, such as the model numbers of ADC, photodetector, controller, signal generator, etc. We have added this information to the revised manuscript. Ref [28] is our latest review paper detailing the power consumption in silicon photonics, which can also help as a good reference. Our photonic BSS technique is agnostic to the mechanism used to shift the ring transmission. Hence, more energy-efficient tuning methods will lead to correspondingly lower power requirements with no change in performance.

Latency can be measured by dividing the length of the overall signal path by the speed of light in the optical waveguide. As shown in Fig. 2a, the received signal mixtures enter this BSS processor at the Mach-Zehnder modulator (MZM, electrical to optical conversion) and output from the balanced photodetector (BPD, for optical to electrical conversion). Fig. 2a has been corrected to show the analog electrical output right after the BPD rather than the ADC. The ADC is part of the statistical analysis required by the BSS algorithm to search for the correct demixing weights. Once the weights are settled, the ADC does not contribute any processing latency to the photonic processor. The latency associated with the chip can be negligible and is mainly from the fiber pigtailed of all the optical components, which have a total length of 3.1 m. The light propagates at the speed of c_0/n , where $c_0 = 299782458$ m/s and the group index of refraction of the fiber is $n = 1.4679$. Thus, the latency can be calculated as 15 ns. We have added this information to the revised manuscript.

[Original] "...It is also worth noting that most of the signal path is in the optical domain, bringing about broadband and flat response and very low latency" (Line 173)

[Revised] "...It is also worth noting that most of the signal path is in the optical domain, bringing about broadband and flat response and very low latency, which is estimated to be 15 ns by dividing the total waveguide length (3 m) by the speed of light in the waveguide ($c_0/n \approx 2 \times 10^8 \text{ m s}^{-1}$).” (Line 174)

[Original] "...The spectra of the four MRRs (at 25 degrees) are shown in Fig. 2c, with the resonance peaks located at 1551.7 nm, 1553.0 nm, 1554.6 nm, and 1555.7 nm..." (Line 179)

[Revised] "...The spectra of the four MRRs (at 25 degrees) are shown in Fig. 2c, with the resonance peaks located at 1551.7 nm, 1553.0 nm, 1554.6 nm, and 1555.7 nm. The waveguide of these MRRs is N-doped which can be efficiently thermally tuned from on to off positions with a power consumption of 10 mW [28]" (Line 180)

6) *Personally, I found hard to follow the paragraph "Photonic BSS algorithm" in the Methods. Although this is not my area of expertise, I believe that few more explanations could benefit the non-expert reader. Moreover, I think it could be useful to know the sampling rate at the oscilloscope: this would allow understanding the convenience of the waveform-agnostic approach.*

Response: We appreciate this critical feedback. We have rewritten the entire "photonic BSS algorithm" in the revised manuscript with more detailed explanations of the algorithm and some additional references for better understanding. We also added two references where detailed pseudo codes can be found for ease of replication.

The oscilloscope we used is the DPO73004sx from Tektronix. BSS does not require Nyquist sampling rates as long as the sampled data points can accurately reflect the variance and kurtosis of the signals. Thus, it is commonly acknowledged that the ADC used for the statistical analysis in BSS can have an arbitrary sampling rate much lower than the signal frequencies. In our experiment, we set the sampling rate to 625 MS/s (the minimal sampling rate) when performing the weight-tuning algorithm. For plotting results, we switched the scope's sampling rate to 100 GS/s (the maximal sampling rate) for clear illustrations. We added these descriptions in the revised manuscript.

[Original] "...Apart from the photonic chip, also included is a BPD for electrical-to-optical (E/O) conversion (DSC-R405ER, Discovery semiconductor), an ADC for signal digitization (DPO73304SX, Tektronix), a computer for statistical analysis and actuating weight commands, and a multi-channel current source for MRR tuning (custom-built as shown in Fig. 2d)." (Line 162)

[Revised] "...Apart from the photonic chip, also included is a BPD for electrical-to-optical (E/O) conversion (Discovery Semiconductor DSC-R405ER), an ADC for signal digitization (Tektronix DPO73304SX; sampling rate: 625 MS s⁻¹-100 Gs s⁻¹), a computer for statistical analysis and actuating

weight commands, and a multi-channel current source for MRR tuning (custom-built as shown in Fig. 2d). The sampling rate of the ADC is set to the minimum of 625 MS s^{-1} for all the tested signals as the BSS method is agnostic to waveform frequencies and is switched to the maximum of 100 GS s^{-1} for recording waveform with the highest definition." (Line 160)

Additional revisions are included in the **Methods**.

7) *The power of the signals is not defined.*

Response: We have added information regarding signal powers in our revised manuscript for every experiment shown in Fig. 3, 4, and 5, as well as additional results in the Supplementary Information. All the signals output by the arbitrary waveform generator (AWG) are set to the maximum amplitude of 1V peak-to-peak. For the wireless system experiment shown in Fig. 3, the received two signal mixtures have much lower powers because of transmission loss, as shown in the spectrum in Fig. 3a-f. The amplitudes of the received signals are about 150 mVpp and 140 mVpp.

[Original] "...Then, the mixtures are received by a 2x2 MIMO antenna (1055-368, 1.7 - 2.5 GHz, Southwest Antennas), with two outputs corresponding to the polarization of 45-degree slant left and 45-degree slant right." (Line 249)

[Revised] "...Then, the mixtures are received by a 2x2 MIMO antenna (Southwest Antennas 1055-368; 1.7 - 2.5 GHz), with two outputs corresponding to the polarization of a 45-degree slant left and a 45-degree slant right. The signal emitted from the transceivers have peak-to-peak voltages of 1 V, and received signals are 150 mV and 140 mV peak-to-peak." (Line 256)

[Original] "...denoting an ill-condition number of 2.26 (according to Equation 2)." (Line 283)

[Revised] "...denoting an ill-condition number of 2.26 (according to Equation 2). The two mixtures are identical in power, that both the peak-to-peak voltages are set to 1 V." (Line 296)

8) *In Fig.2f, not all the red points show >9 bits. How have you calculated the precision? Is that an average of the collected points?*

Response: The precision is calculated by the standard deviation of all the tested points shown in Fig. 2e and f, which is why some tested results lay outside the 9-bit scope. We performed this accuracy test at each point three times, as shown in Fig. 2e. Thus, the total number of test points is $9 \times 9 \times 3 = 243$.

Quantitatively, 161 among the 243 points show accuracy better than 9 bits (inside the solid circle). We have provided this description in the revised manuscript to avoid any confusion.

[Original] "...A 9.0-bit of precision resulted from the dithering control and 6.7-bit for the control without the dithering." (Fig. 2 caption)

[Revised] "...A 9.0-bit of precision resulted from the dithering control and 6.7-bit for the control without the dithering. **The precision is calculated by the standard deviation of the error. Quantitatively, there are 161 among the total 243 tested points inside the 9-bit bound (the solid circle).**" (Fig. 2 caption)

9) *In the "Results" paragraph, the reported model of generic mixing considers two signals. In the case study, the target is to isolate one signal from an interferer. Is it correct to consider the interferer as a signal, knowing that it could be the random summation of several transmission channels with different frequencies, bandwidths, statistics?*

Response: Thank you for raising this question. We believe the interferer can be considered as a signal. Our BSS algorithm is based on the central limit theorem, which assumes the original signals to be independent and uncorrelated from each other. In the experiment shown in Fig. 3, the interferer is an instantaneous broadband signal representing a wideband jammer signal. This jammer signal is uncorrelated with the signal of interest and transmitted by a dedicated antenna. Thus, the interferer can be considered a standalone signal. This is the reason that our photonic BSS system can recover this interferer the same way as the signal of interest, as testified in Fig. 3c. In fact, this BSS approach can be used to recover every original signal from their mixtures, which is not only limited to isolating the signal of interest from an interferer. This BSS approach can also separate the interferer from the signal if the noise-free interferer is demanded.

In some applications, as mentioned by the reviewer, the interferer comes from the random summation of several transmission channels. If the interferer contains no statistical information or properties about the signal of interest, the interferer can still be regarded as a standalone signal that is uncorrelated with the signal of interest. Thus, our BSS photonic system can perform the separation on the mixtures of this interferer and the signal of interest. Suppose the interferer is correlated with the signal of interest, which means the interferer is a mixture of the signal of interest and other signal components. In that case, the interferer can no longer be considered a standalone signal; and cannot be solved by our BSS setup since the central limit theorem does not apply to correlated signals.

10) *SIR at line 111 is not defined (the definition of the acronym comes later in the paper). PC and IC at lines 350 and 356 are not defined neither. In "Photonic hardware implementation", the external MZMs are not described.*

Response: We have moved the definition of the SIR, which stands for signal-to-interference ratio, to the introduction, around line 111. We also added the definitions of PC and IC, which are principal components and independent components, respectively, at around lines 350 and 356. Also, the definition of MZMs is added in the "Photonic hardware implementation" section. We have double-checked the revised manuscript to ensure all the acronyms have been defined.

[Original] "...In terms of performance, our setup fully realizes the “blindness” agility by achieving a bandwidth of up to 19.2 GHz and SIR of more than 40 dB in some cases." (Line 111)

[Revised] "...In terms of performance, our setup fully realizes the “blindness” agility by achieving a processing bandwidth of up to 19.2 GHz and **signal-to-interference ratio (SIR)** of more than 40 dB in some cases." (Line 109)

[Original] "...Firstly, photonic PCA updates the MRR weights to converge at the target PC vectors by maximizing the variance of weighted addition output $E(y_i^2)$." (Line 350)

[Revised] "...**Firstly, PCA is to find the principal components (PC) from the received mixtures and construct a whitening matrix.**"(Line 368)

[Original] "...Then, photonic ICA utilizes this whitening matrix V and updates the MRR weight vector in the whitened subspace $\$w_i V(i = 1, \dots, n)\$$ to converge at the target IC vectors by maximizing the absolute relative kurtosis of the output," (Line 354)

[Revised] "... **Secondly, the ICA step follows a similar search procedure and finds the vector (denoted as the first independent component (IC)) corresponding to the maximal kurtosis.** " (Line 381)

[Original] "...The signal path starts from the MZM and ends at the scope" (Line 171)

[Revised] "...The signal path starts from the **Mach-Zehnder Modulator (MZM)** and ends at the scope" (Line 170)

11) The position along the text of Eq. (3) to (5) is strange. It would be better having the equations in the middle of the phases that cite them. Moreover, I found difficult following the calculations (I admit I am not an expert of these technical details).

Response: We agree with the reviewer that the derivation of the equations can be more precise. We have added an intermediary step to these equations and a few sentences detailing how each equation is obtained. More importantly, we moved the equations to the closest position where the main text cites them. Hopefully, these adjustments will make it easier for the readers to follow.

12) *At line 283, the mixing of the two signals is given, I believe one of the two should be M2...*

Response: The reviewer is correct. The second equation in line 283 should be associated with M_2 . We double-checked the revised manuscript to ensure it is clear of such typos.

[Original] "...the mixing can be expressed as $M_1 = 0.8 \times S_1 + 0.2 \times S_2$ and $M_1 = 0.2 \times S_1 + 0.8 \times S_2$," (Line 280)

[Revised] "...the mixing can be expressed as $M_1 = 0.8 \times S_1 + 0.2 \times S_2$ and $M_2 = 0.2 \times S_1 + 0.8 \times S_2$ " (Line 313)

REVIEWERS' COMMENTS

Reviewer #1 (Remarks to the Author):

The authors have well addressed my previous concerns. I think this manuscript can be considered for publication.

Reviewer #2 (Remarks to the Author):

The authors have addressed the comments of the reviewer very clearly and convincingly.

Point-by-point reply to reviewers' comments:

Reviewer 1

The authors have well addressed my previous concerns. I think this manuscript can be considered for publication.

Response: It's our pleasure to hear that all the reviewer's concerns are addressed. We appreciate the reviewer for the recommendation for publication.

Reviewer 2

The authors have addressed the comments of the reviewer very clearly and convincingly.

Response: We are pleased to see the satisfaction from the reviewer in terms of our revised manuscript. We would like to appreciate the reviewer's contribution sincerely.